# Targeting AHR Increases Pancreatic Cancer Cell Sensitivity to Gemcitabine through the ELAVL1-DCK Pathway

**DOI:** 10.3390/ijms241713155

**Published:** 2023-08-24

**Authors:** Darius Stukas, Aldona Jasukaitiene, Arenida Bartkeviciene, Jason Matthews, Toivo Maimets, Indrek Teino, Kristaps Jaudzems, Antanas Gulbinas, Zilvinas Dambrauskas

**Affiliations:** 1Surgical Gastroenterology Laboratory, Institute for Digestive Research, Lithuanian University of Health Sciences, Eiveniu 4, 50103 Kaunas, Lithuania; aldona.jasukaitiene@lsmu.lt (A.J.); arenida.bartkeviciene@lsmu.lt (A.B.); antanas.gulbinas@lsmu.lt (A.G.); zilvinas.dambrauskas@lsmu.lt (Z.D.); 2Department of Nutrition, Institute of Basic Medical Sciences, University of Oslo, 1046 Blindern, 0317 Oslo, Norway; jason.matthews@medisin.uio.no; 3Department of Pharmacology and Toxicology, University of Toronto, Toronto, ON M5S 1A8, Canada; 4Institute of Molecular and Cell Biology, University of Tartu, Riia 23, 51010 Tartu, Estonia; toivo.maimets@ut.ee (T.M.); indrek.teino@ut.ee (I.T.); 5Latvian Institute of Organic Synthesis, Aizkraukles 21, LV-1006 Riga, Latvia; kristaps.jaudzems@osi.lv

**Keywords:** AHR, ELAVL1, PDAC, gemcitabine, HMOX1, DCK

## Abstract

The aryl hydrocarbon receptor (AHR) is a transcription factor that is commonly upregulated in pancreatic ductal adenocarcinoma (PDAC). AHR hinders the shuttling of human antigen R (ELAVL1) from the nucleus to the cytoplasm, where it stabilises its target messenger RNAs (mRNAs) and enhances protein expression. Among these target mRNAs are those induced by gemcitabine. Increased AHR expression leads to the sequestration of ELAVL1 in the nucleus, resulting in chemoresistance. This study aimed to investigate the interaction between AHR and ELAVL1 in the pathogenesis of PDAC in vitro. *AHR* and *ELAVL1* genes were silenced by siRNA transfection. The RNA and protein were extracted for quantitative real-time polymerase chain reaction (qRT-PCR) and Western blot (WB) analysis. Direct binding between the ELAVL1 protein and *AHR* mRNA was examined through immunoprecipitation (IP) assay. Cell viability, clonogenicity, and migration assays were performed. Our study revealed that both AHR and ELAVL1 inter-regulate each other, while also having a role in cell proliferation, migration, and chemoresistance in PDAC cell lines. Notably, both proteins function through distinct mechanisms. The silencing of ELAVL1 disrupts the stability of its target mRNAs, resulting in the decreased expression of numerous cytoprotective proteins. In contrast, the silencing of *AHR* diminishes cell migration and proliferation and enhances cell sensitivity to gemcitabine through the AHR-ELAVL1-deoxycytidine kinase (DCK) molecular pathway. In conclusion, AHR and ELAVL1 interaction can form a negative feedback loop. By inhibiting AHR expression, PDAC cells become more susceptible to gemcitabine through the ELAVL1-DCK pathway.

## 1. Introduction

Pancreatic cancer (PC) is a devastating disease that has a less than 10% five-year survival rate [1,2], with pancreatic cancer ductal adenocarcinoma (PDAC) being the most common in pancreatic malignancies. Future projections show that the death rate from PC will increase by 1.76% annually [3] and could be one of the leading forms of cancer by mortality by 2030 [4,5]. Curative surgery followed by chemotherapy remains the recommended treatment for resectable pancreatic tumours; however, due to PC being diagnosed mainly in advanced stages, curative surgery is only available in 15–20% of cases [6]. In cases with advanced or metastatic PC, gemcitabine (GEM) remains one of the first-line drugs for chemotherapy, although other multidrug regimens are also widely used (e.g., FOLFIRINOX or FOLFOXIRI) [7]. Due to poor future projections, low diagnostic statistics, low survival, and limited treatment options, it is imperative to search for new or improve existing treatment of PC.

One of the biggest problems in PC treatment is its heterogeneity and resistance to first-line drugs. Resistance to chemotherapy can be innate (germinal genetic makeup) or acquired (mutational) [8], with the latter being common in PC. A common difficulty of PC treatment is that it becomes chemoresistant weeks after starting treatment, which could also be attributed to alterations in post-transcriptional gene regulation. The human antigen R (ELAVL1) protein is an RNA-binding protein that binds mRNAs and increases their stability. There are numerous targets of ELAVL1 which are involved in protective mechanisms (Heme oxygenase 1 [9] or Cyclooxygenase-2 [10]), proliferation, differentiation, apoptosis, and senescence of the cell [11,12]. One of the enzymes regulated by ELAVL1 is deoxycytidine kinase (DCK)—an enzyme responsible for initiating the process of GEM phosphorylation, which leads to its incorporation into the DNA and subsequently arrested cell cycle. The overexpression of ELAVL1 upregulates and silencing downregulates the expression of DCK in PC [13,14]. However, to successfully regulate the expression of DCK and the subsequent response to gemcitabine, the ELAVL1 protein has to be located in the cytoplasm of the cell. The concentration of cytoplasmic ELAVL1 is positively associated with a response to gemcitabine and survival of the patients [10]. DNA damage, such as the effect of gemcitabine, works as a stimulus for ELAVL1 to translocate from the nucleus to the cytoplasm [15]; however, even after such stimuli, ELAVL1 can be sequestered into the nucleus. One of the reasons for this sequestering of ELAVL1 could be the aryl hydrocarbon receptor (AHR) [16]. Normally, AHR is responsible for cell defence and immune system regulation [17,18]. Increases in AHR expression are seen in autoimmune diseases and various forms of cancer [19], and AHR is frequently overexpressed in pancreatic cancer [20,21,22]. In cases with AHR overexpression, ELAVL1 could be sequestered in the nucleus, which would increase the chemoresistance of the cells.

Therefore, we hypothesise that the modulation of AHR and ELAVL1 expression can decrease the chemoresistance of PDAC cells in vitro.

## 2. Results

### 2.1. Modulation and Relationship of AHR and ELAVL1 in PDAC Cell Lines

Human PDAC cell lines (BxPC-3, Su.86.86) were transfected with small interfering RNAs (siRNAs): siAHR or siELAVL1 for 24 h. Western blot analysis revealed that the *AHR* and *ELAVL1* genes were silenced almost completely after 24 h in both lines. In BxPC-3, siAHR decreased *AHR* mRNA to 20% and the protein was not detected; in Su.86.86, *AHR* mRNA decreased to 29% and the protein decreased to 2% compared with nontreated control cells.

When silencing *ELAVL1* in BxPC-3 cells, the *ELAVL1* mRNA levels decreased to 31%, and the ELAVL1 protein levels decreased to 38% compared with nontreated control cells. Similarly, in Su.86.86 cells, the *ELAVL1* mRNA levels decreased to 9%, and the ELAVL1 protein levels decreased to 15% compared with nontreated control cells. These results indicate successful silencing of both the target mRNA and the subsequent protein expression for ELAVL1 (see Figure 1).

Moreover, *AHR* silencing increased ELAVL1 mRNA in both cell lines (BxPC3—1.4-fold; Su.86.86—1.2-fold) (Figure 1). The same increase was seen in protein levels (BxPC-3 ELAVL1 protein increased 1.83-fold and Su.86.86 1.2-fold).

*ELAVL1* silencing caused a decrease in *AHR* mRNA (in BxPC-3 *AHR* mRNA decreased significantly to 76% and in Su.86.86 to 77%) as well as in protein (in BxPC-3 AHR protein was absent and in Su.86.86 decreased to 73%).

These findings suggest a mutual influence and potential regulatory relationship between AHR and ELAVL1. As ELAVL1 primarily exerts its activity in the cytoplasm, it was important to investigate the mRNA and protein expression levels of its downstream targets, specifically DCK and HMOX1.

The silencing of *AHR* increased Heme oxygenase 1 (*HMOX1*) mRNA (4.45-fold) and protein (1.4-fold) in BxPC-3 cells; however, Su.86.86 showed no changes in either mRNA or protein of HMOX1 (Figure 2). SiAHR also increased *DCK* mRNA (1.6-fold) and protein slightly (1.1-fold) in the BxPC-3 cell line; however, it did not change *DCK* mRNA and even decreased protein levels (to 56%) in the Su.86.86 cell line.

The silencing of *ELAVL1* decreased *HMOX1* mRNA (BxPC-3—79%; Su.86.86—81%) and protein (BxPC-3—88%; Su.86.86—57%) in both cell lines. *DCK* mRNA did not change in either cell line; however, the DCK protein decreased in both (BxPC-3—65%; Su.86.86—57%).

These results show that ELAVL1 pathway genes and proteins somewhat correspond to changes in AHR and ELAVL1 expression levels; however, it is more noticeable for protein level and differs between cell lines.

### 2.2. ELAVL1-Mediated Post-Transcriptional Regulation of AHR as Demonstrated by Immunoprecipitation

Since a relationship between AHR and ELAVL1 can be seen in mRNA and protein expression levels, the IP assay was used to determine a direct link between *AHR* mRNA and the ELAVL1 protein. ELAVL1 is known to bind to various mRNAs, including *DCK* [13] and *HMOX1* [9]. The *AHR* gene has nine adenylate uridylate (AU)-rich elements (AREs) (ATTTA), which is used as a binding motif for ELAVL1 protein [23]. The qRT-PCR results of IP showed that the ELAVL1 protein binds *AHR* mRNA (Figure 3a). As a control for the assay, *GAPDH* mRNA was tested as a negative control, which the ELAVL1 protein did not bind to, and *HMOX1* mRNA as a known target for a positive control (Figure 3b). *AHR* mRNA sequence analysis, together with our experimental data, indicate that the ELAVL1 protein binds *AHR* mRNA and thereby potentially modulates AHR translation. Western blot assay was used as a control of the IP assay (Figure 3c).

### 2.3. Gemcitabine IC50 Dose Determination

The metabolic activity of PDAC cells was determined by MTT assay. IC50 doses of GEM were determined separately for both cell lines: Su.86.86—36.91 ± 1.40 nM, BxPC-3—26.67 ± 1.69 nM (Figure 4a,b).

### 2.4. Cellular Localisation Changes in ELAVL1 in Response to AHR Silencing and/or Gemcitabine Treatment

Cellular localisation shifts in ELAVL1 and AHR protein after the silencing of ELAVL1 or AHR genes was elucidated by immunocytochemistry (ICC).

AHR mainly resides in the cytoplasm of both PDAC cell lines. The shift of AHR protein was not noticeable in either cell line when silencing ELAVL1 and/or treating with GEM (photos not included).

The ELAVL1 protein mainly resides in the nucleus of both PDAC cell lines (Figure 5a,e). When treating cells with GEM, the concentration of cytoplasmic ELAVL1 increases (Figure 5b,f), showing that GEM acts as a signal for ELAVL1 to shift its localisation. When silencing AHR, the concentration of cytoplasmic ELAVL1 increases (Figure 5c,g), showing that AHR can be involved in sequestering ELAVL1 in the nucleus. The joint effect of *AHR* silencing and GEM treatment increased cytoplasmic ELAVL1 concentrations even more than separately, indicating that the silencing of *AHR* could be beneficial to ELAVL1 translocation in response to GEM treatment (Figure 5d,h).

### 2.5. AHR and ELAVL1 Modulation Influences the Chemoresistance of PDAC Cells

GEM treatment significantly increased *AHR* mRNA (BxPC-3—4.3-fold; Su.86.86—2.4-fold) but decreased protein levels in both cell lines (BxPC-3 to 27%; Su.86.86 to 38%) (Figure 6). Treatment also increased *ELAVL1* mRNA (BxPC-3—1.34-fold; Su.86.86—1.22-fold) and protein (BxPC-3—2.1-fold; Su.86.86—1.1-fold).

When silencing *AHR* prior to GEM treatment, it increased BxPC-3 *ELAVL1* mRNA (1.46-fold) and protein (2.6-fold); however, Su.86.86 *ELAVL1* mRNA did not change, and protein decreased to 74%. Silencing *ELAVL1* prior to GEM treatment increased *AHR* mRNA in both cell lines (BxPC-3—2.2-fold; Su.86.86—1.4-fold) but greatly decreased AHR protein (BxPC-3 to 12%; Su.86.86 to 38%).

These findings highlight the differences observed between mRNA levels and protein expression levels, as well as variations across different cell lines. These differences suggest the involvement of post-transcriptional regulatory mechanisms in the cellular response to gemcitabine. To gain a deeper understanding of these distinctions, we conducted an analysis of *ELAVL1* pathway genes, specifically *HMOX1* and *DCK*.

GEM greatly increased *HMOX1* mRNA (BxPC-3—7.2-fold; Su.86.86—2.7-fold); however, an increase in protein was much less noticeable (BxPC-3—1.2-fold, Su.86.86—1.1-fold), as well as *DCK* mRNA (BxPC-3—2-fold, Su.86.86—2.3-fold) and protein (BxPC-3—2.5-fold; Su.86.86—1.3-fold) (Figure 7).

When silencing *AHR* prior to GEM treatment, *HMOX1* mRNA increased (BxPC-3—13.6-fold; Su.86.86—1.5-fold); however, the HMOX1 protein increased in BxPC-3 (1.5-fold) and decreased in Su.86.86 (to 75%). A similar pattern was also seen with *DCK* mRNA and protein levels. *DCK* mRNA increased in both cell lines (BxPC-3—2.1-fold; Su.86.86—1.3-fold); however, protein levels only increased in BxPC-3 (2.5-fold) and did not change in Su.86.86.

The silencing of *ELAVL1* prior to GEM treatment increased *HMOX1* mRNA (BxPC-3—3.1-fold; Su.86.86—1.7-fold) and protein (BxPC-3—1.1-fold; Su.86.86—1.4-fold). *DCK* mRNA was also increased (BxPC-3—2.1-fold; Su.86.86—1.4-fold), as well as protein levels (BxPC-3—2.6-fold; Su.86.86—2.1-fold).

The observed changes indicate that both AHR and ELAVL1 genes and proteins, as well as the ELAVL1 pathway genes *HMOX1* and *DCK*, respond to gemcitabine (GEM) treatment. However, notable differences exist between mRNA and protein expression levels of AHR, suggesting the involvement of post-transcriptional regulation mechanisms that affect the translation of *AHR* mRNA into protein. Furthermore, significant differences were observed between the cell lines, indicating the presence of additional mechanisms that contribute to the observed variations.

To determine the effect of *AHR* or *ELAVL1* gene silencing on cell chemoresistance, *AHR* or *ELAVL1* genes were silenced for 24 h by transfection with siAHR or siELAVL1. After silencing, the cells were treated with an IC50 dose of GEM for 48 h. The metabolic activity of all groups was compared with the control group (100%) of nontreated cells for GEM alone or siControl for transfected cells. GEM alone significantly reduced cell viability to 57% for the BxPC-3 cell line and 48.4% for the Su.86.86 cell line. There was no significant effect on BxPC-3 or Su.86.86 cell viability after the silencing of *AHR* or *ELAVL1*. However, the silencing of *AHR* together with GEM significantly decreased the viability of both cell lines when compared with GEM alone (BxPC-3 to 25.8% and Su.86.86 to 23.1% cell viability). In BxPC-3, the silencing of *ELAVL1* together with GEM did not have a significant effect on cell viability when compared with GEM alone; however, in Su.86.86, the combined effect of siELAVL1 and GEM when compared with GEM alone significantly decreased cell viability to 23.1% (Figure 8a,b).

The ability of cell lines to form colonies was measured by clonogenicity assay (Figure 9). The effect of gemcitabine was more detrimental to long-term than short-term (MTT) cell survival. IC50 doses (MTT) of GEM significantly decreased cell colony formation to 18.3% compared with the nontreated control in the BxPC-3 cell line and to 30.3% in the Su.86.86 cell line. The silencing of *AHR* also had a more detrimental effect on long-term than short-term survival. Both cell lines had a significant decrease in colony formation (19.3% formation in the BxPC-3 cell line and 27.7% formation in the Su.86.86 cell line compared with the nontreated control). The silencing of *ELAVL1* had a significant effect on BxPC-3 colony formation (colony formation decreased to 74.7%) but had no effect on Su.86.86 cell line colony formation. The silencing of *AHR* significantly increased the effect of GEM (BxPC-3 to 3.7% and Su.86.86 to 11.3%); however, due to severe effects on both GEM and siAHR, it is unclear whether the effect is due to increased sensitivity to GEM or the additive effect of siAHR and GEM. The silencing of *ELAVL1* did not significantly change the effects of GEM on either cell line.

Migratory ability was measured by the wound healing (scratch) assay (Figure 10). After 24 h, the BxPC-3 cell line (Figure 10a,c) showed significantly lower migratory abilities in response to GEM (41.2% wound closure compared with control cells who had 98.29% wound closure). The silencing of *AHR* significantly reduced BxPC-3 cell line migratory abilities, and 36.2% of the wound was closed after 24 h. On the other hand, the silencing of *ELAVL1* had no significant effect on the migration of BxPC-3 cells. The silencing of *AHR* or *ELAVL1* and treatment with GEM had a significant effect when compared with controls (siAHR+GEM—16.5% and siELAVL1+GEM—19.7% of the wound closed); however, neither siAHR nor siELAVL1 increased GEM effect on migratory abilities. Su.86.86 (Figure 10b,d) had a lower migratory ability overall and no significant decrease in migratory ability after any of the effects compared with the control. The silencing of *AHR* seems to have a significant effect on cell migratory abilities; however, it was noticeable only in highly migratory cells (BxPC-3), and it did not increase the effects of GEM on cell migration.

## 3. Discussion

Chemoresistance of pancreatic cancer is a major problem limiting the success of chemotherapeutic treatments in patients. There are plenty of suggested mechanisms that cause PC resistance to chemotherapy, including common genetic mutations such as those in the *KRAS* [24,25], *MUC4* [26], and *TP53* [27] genes. MicroRNAs such as *miR-106b* [28], *miR-181b* [29], *miR-21* [30], and *miR-29a* [31] have also been implicated in the chemoresistance of PC. Even the overexpression of ribonucleotide reductase subunit M1 [32] or heat shock protein 27 [33] and many others are thought to be involved in one way or another. One of the most common proteins implicated in PC resistance to gemcitabine is the RNA-binding protein ELAVL1. Its cytoplasmic concentration is directly associated with the longer survival of PC patients [10]. AHR, which is often overexpressed in PC [20,21,22], is thought to be involved in blocking ELAVL1 from leaving the nucleus [17] and in turn stabilising its target mRNAs. One of the targets of ELAVL1 which is also implicated in gemcitabine resistance is DCK [34,35,36]. This is an enzyme that starts the activation of gemcitabine by phosphorylation; however, it is usually inactive in gemcitabine-resistant cells [37]. Therefore, in cells with upregulated AHR, ELAVL1 would be sequestered in the nucleus, which would decrease DCK protein concentrations and, in turn, contribute to gemcitabine resistance.

Our results show that *AHR* mRNA is a direct target for the ELAVL1 protein, and this interaction stabilises *AHR* mRNA, thus possibly increasing protein synthesis. AHR has been shown to sequester ELAVL1 in the nucleus [16], which in turn would block it from stabilising the mRNAs of various proteins, including *AHR*. By silencing *AHR* mRNA and in turn protein synthesis, we were able to show an increase in ELAVL1 expression as well as its localisation shift towards cytoplasm, which agrees with previous studies. However, the results of the expression of ELAVL1 pathway genes and proteins are conflicting, and both cell lines had different changes in some cases, showing that the effect of this modulation is hard to predict and can be different depending on the cell characteristics. By silencing *ELAVL1*, we showed that *AHR* mRNA and protein levels decrease, proving again that AHR can be a post-transcriptional target of ELAVL1. This complex interaction between AHR and ELAVL1 shows that they can alter the expression of each other by forming a negative feedback loop, which has never been shown before.

Due to the fact that AHR is a transcription factor and ELAVL1 is a post-transcriptional gene expression regulator, this interaction and its changes involve many different cell mechanisms involved in cytoprotection, migration, overall viability of the cell, and most importantly to PC treatment chemoresistance. Our results show that by lowering AHR or ELAVL1 expression, the cells become more susceptible to gemcitabine; however, the mechanisms most likely differ. The silencing of *AHR* greatly reduces cell migration and colony formation and might ease ELAVL1 translocation from the nucleus to the cytoplasm. However, different responses of ELAVL1 pathway genes and differences between cell lines show that there are more mechanisms involved in this regulation, warranting further investigation into the relationship between AHR and ELAVL1. Increased AHR expression can be seen in many different malignancies, for example, lymphoma [38] and leukaemia [39], as well as breast [40], kidney [41], gastrointestinal [42], and pancreatic cancers [20,21]. Since AHR is often overexpressed in cancer, it is frequently targeted in the hope of increasing the effectiveness of cancer treatment in various malignancies [20,40,43], and in this case, it could be targeted in the hopes of increasing gemcitabine efficiency in PC treatment.

The silencing of *ELAVL1* has a much lesser effect on cell migration or colony formation than *AHR*; however, our results show that it decreases *HMOX1* mRNA levels and DCK and HMOX1 protein levels. This lowers the protective mechanisms of the cell but can also increase the chemoresistance of the cell. Lowering ELAVL1 was also shown to decrease the synthesis of other protective proteins, such as COX-2 [44] or IFN-β [45], making the cell more susceptible to stress. ELAVL1 was shown to be both a positive and negative marker for various malignancies. Its interaction with various microRNAs has been shown to be a negative factor of ovarian cancer [46], prostate cancer [47], and other malignancies. It was shown to activate the *MAPK* and *JNK* signalling pathways and cause breast cancer resistance to tamoxifen [48]; its cytoplasmic concentration was shown to be a negative sign for the treatment of invasive breast cancer [49]. In pancreatic cancer, ELAVL1 was shown to regulate apoptosis through the IAP1 and IAP2 proteins [50]. It can also stop the cell cycle at the G2/M phase, allowing the cell to repair DNA damage, thus avoiding apoptosis [15]. The silencing of *ELAVL1* was also shown to increase the response to chemotherapy, although it was attributed to a decrease in cytoprotective proteins rather than a direct influence on gemcitabine activation [44]. However, in PC, more often than not, increased ELAVL1 concentrations in the cell cytoplasm are considered a positive sign, not only in terms of response to gemcitabine but also in overall patient survival [10,13,14,51].

Since ELAVL1 can be both a good and bad marker for chemotherapy resistance, it is imperative to understand how it works and find ways to utilise it in a way that would not cause any harm. Our findings suggest a mechanism (Figure 11) by which PDAC cells might be able to have increased resistance to gemcitabine through the ELAVL1-DKC pathway and highlights AHR as a target molecule to negate that resistance. Following the cellular uptake of gemcitabine, DCK starts its phosphorylation, which leads to cell cycle arrest and ideally cancer cell death (Figure 11(1–3)). ELAVL1 is a regulator of DCK and translocates to the cytoplasm following DNA damage, where it stabilises target mRNAs and increases their protein synthesis. In turn, increased DCK synthesis further strengthens the cytotoxic effect of gemcitabine (Figure 11(5,6)). However, at the same time, ELAVL1 stabilises the mRNAs of cytoprotective proteins and AHR, which in turn sequesters ELAVL1 in the nucleus, subsequently causing a negative feedback loop (Figure 11(5,7,8)). In cases of AHR overexpression in PC, ELAVL1 would be further sequestered in the nucleus, blocking it from stabilising the DCK protein and causing resistance to gemcitabine. 

Overall, targeting AHR and decreasing its expression would ease the shuttling of ELAVL1 to the cytoplasm, which in turn would decrease cell resistance to gemcitabine and, at the same time, decrease cell colony formation and migration capabilities. Notably, the increase in ELAVL1 activity can increase cell cytoprotective mechanisms through the stabilisation of proteins such as HMOX1, although this effect depends on the cell characteristics according to our results, warranting further studies into the mechanisms involved.

Both AHR and ELAVL1 are involved in various molecular mechanisms. ELAVL1 is known to be involved in various cell signalling pathways, such as *MAPK* and *JNK* [48], as well as ferroptosis activation [52]. It is also involved in various systems, such as innate immune barriers in infants [53] and the overall immune system [45,53]. Similarly, AHR is also involved in various cell signalling pathways, such as epidermal growth factor family pathways [54] and xenobiotic metabolism [55]. It is also highly involved in the immune response systems [56,57]. The wide range of pathways AHR and ELAVL1 are involved with includes numerous interaction partners that could be regulated by modulating the AHR-ELAVL1 pathway, thus not limiting it to cytoprotection or gemcitabine metabolism.

Both AHR and ELAVL1 are involved in the immune system of the body. This poses a limitation to in vitro experiments due to absence of immune cells. Nonetheless, this novel mechanism provides a stepping stone for combating gemcitabine resistance in PC and should be investigated in in vivo setting. This would further elucidate the relationship of AHR and ELAVL1 in a setting with various other systems with which both AHR and ELAVL1 might interact.

There are some limitations to our study. Only lipofectamine-mediated siRNA transfection was used, which would prove to be difficult to use for patients, and so further studies with protein inhibitors would broaden the knowledge of the AHR-ELAVL1-DCK pathway. Also, only two cell lines were used, which had some different responses to AHR and ELAVL1 modulation; so, further investigation into the mechanisms causing such differences and the stratification of cancer subtypes that would benefit most from such modifications are required. Lastly, only in vitro experiments with cell cultures were performed; so, further experiments in vivo would be necessary to prove this relationship.

## 4. Materials and Methods

### 4.1. PDAC Cell Lines and Growing Conditions

Two human pancreatic ductal adenocarcinoma cell lines, BxPc-3 and Su.86.86, were used for analysis. The BxPC-3 and Su.86.86 cell lines were a gift from the European Pancreas Centre (Heidelberg, Germany). Both cell lines were grown in RPMI medium (Gibco, Life Technologies Limited, Paisley, UK) with 10% FBS (Gibco Life Technologies Limited, Paisley, UK) and 1% penicillin/streptomycin solution (Gibco). Cells were grown in monolayers in sterile flasks/plates in an incubator, which maintains a moist temperature of 37 °C with a 5% CO_2_-enriched environment.

### 4.2. Gemcitabine Treatment of Cells and IC50 Measurement

Cells were seeded in 96-well cell culture plates (BxPC-3—2.5 × 10^3^ cells/well; Su.86.86 1.2 × 10^3^ cells/well) and treated with varying concentrations of gemcitabine (GEM) (Fluorochem, Glossop, UK) ranging from 1–10,000 nM by logarithmic dilution to test the half-maximal inhibitory concentration (IC50) of GEM. Treated cells were maintained at 37 °C in a 5% CO_2_ incubator for 48 h. To measure the proportion of metabolic activity, the MTT (3-(4,5-dimethylthiazol-2-yl)-2,5 diphenyltetrazolium bromide) metabolism method was used (see ‘MTT metabolic activity assay’). No less than 4 replicates of the experiment were carried out.

### 4.3. MTT Metabolic Activity Assay

Cell viability was assessed by MTT (Invitrogen, Carlsbad, CA, USA) assay. After treatment, MTT was added to a concentration of 0.5 mg/mL, and the cells were incubated for 4 h at 37 °C. After 4 h of incubation, the medium with MTT was removed and the remaining formazan product was dissolved with DMSO (dimethyl sulfoxide) (Carl Roth GmbH, Karlsruhe, Germany) by agitation in the spectrophotometer for 60 s. The absorbance was measured with the spectrophotometer (TheSunrise (Software v7.1.), Tecan, Grodig, Austria) at a wavelength of 570 nm and reference of 620 nm.

### 4.4. Transfection

siELAVL1, siAHR, and negative control siRNA were purchased from Ambion (Waltham, MA, USA). Transfection was performed in 96, 6-well plates or 25 cm^2^ flasks. Lipofectamine 2000 (Gibco, Life Technologies Limited, Paisley, UK) was used according to the manufacturer’s instructions for all transfections with RPMI medium. All MTT experiments included two groups of control cells: untreated control and a control treated with an siRNA negative control. Four replicates of the experiment were carried out. Silencing efficiency after 24–72 h was evaluated by Western blot (WB) analysis.

### 4.5. RNA Extraction and Quantitative Real-Time Polymerase Chain Reaction (qRT-PCR)

Total RNA extraction was performed from cultured cells using the RNA extraction kit (Abbexa, Cambridge, UK) according to the manufacturer’s protocol. Purified RNA was quantified and assessed for purity by UV spectrophotometry (NanoDrop 2000 (Software v.1.4.2), ThermoFisher Scientific, Waltham, MA, USA). cDNA was generated from 2 μg of RNA with High-Capacity cDNA Reverse Transcription Kit (Applied Biosystems, Waltham, MA, USA). The amplification of specific RNA was performed in a 20 μL reaction mixture containing 2 μL of cDNA template, 1X PCR master mix, and the primers. The PCR primers used for detection of *ELAVL1* (Hs00171039_m1), *AHR* (Hs00169233), *HMOX1* (Hs01110250_m1), and housekeeping gene *GAPDH* (Hs02758991_g1) were from Applied Biosystems. Three replicates of the experiment were carried out.

### 4.6. Western Blot Analysis

Whole cells were lysed using RIPA lysis buffer with protease inhibitors (Roche, Basel, Switzerland) and centrifuged at 10,000× *g* for 10 min at 4 °C. The supernatants were assayed for protein concentration with BCA protein assay kit (Thermo Scientific, Waltham, MA, USA). Protein samples were heated at 97 °C for 5 min before loading, and 25–50 µg of the samples was subjected to 4–12% sodium dodecyl sulfate-polyacrylamide gel electrophoresis (SDS-PAGE), then transferred to poly-vinylidene fluoride (PVDF) membranes 40 min 30 V. Next, membranes were blocked with a 5% skimmed milk blocking buffer for 60 min at room temperature. Membranes were then incubated for 1.5 h at room temperature or overnight at 4°C with primary antibodies. The following primary antibodies were used: 1:100 mouse monoclonal anti-ELAVL1 (LsBio, Lynnwood, WA, USA; Ls-C7451); 1:1000 mouse monoclonal anti-AHR (ThermoFisher Scientific, Waltham, MA, USA; MA1-514); 1:2000 rabbit monoclonal anti-HO-1 (Abcam; ab68477); 1:3000 mouse monoclonal anti-GAPDH (ThermoFisher Scientific; AM4300); and 1:2000 mouse monoclonal anti-DCK (ThermoFisher Scientific; MA5-25502). The membranes were washed with 1X Tris-Buffered Saline, 0.1% Tween 20 Detergent (TBST) antibody washing buffer or antibody wash buffer (Invitrogen, Carlsbad, CA, USA) and incubated in the appropriate peroxidase-conjugated secondary antibody solution (Invitrogen, Carlsbad, CA, USA) for 30 min or in horseradish-peroxidase-conjugated secondary antibody solution (LSbio) for 1 h. After that, membranes were washed again with TBST antibody washing buffer or antibody wash buffer (Invitrogen, Carlsbad, CA, USA) and incubated with chemiluminescence substrate (Invitrogen, Carlsbad, CA, USA) or West Pico Stable peroxidase buffer + luminol enhancer (Thermo Scientific) for 5 min. Results were analysed with a documenting system (Biorad, Hercules, CA, USA). Three replicates of the experiment were carried out; however, only one is being shown as a representative replicate.

### 4.7. Immunocytochemistry

Cells were cultivated on chamber slides for 96 h either with or without treatment. A mixture of 96% ethanol with 5% glacial acetic acid was used for fixation and 0.5% Triton X-100 in PBS- for permeabilisation. Cells were subsequently incubated with 1:250 primary mouse monoclonal ELAVL1 antibody (LSBio) or 1:500 primary mouse monoclonal AHR antibody (Thermo Scientific, Waltham, MA, USA) and 1:2000 secondary antibody-Alexa Fluor 488 Goat Antimouse IgG (H+L) (Invitrogen, Carlsbad, CA, USA) and washed with PBS followed by washing with nuclease-free water. Slides were then mounted with ProLong Diamond Antifade Mountant with DAPI (Invitrogen, Carlsbad, CA, USA) for cell nuclei staining and analysed with Olympus IX71 fluorescent microscope (Olympus Corporation, Tokyo, Japan). Three replicates of the experiment were carried out; however, only one is being shown as a representative replicate.

### 4.8. Clonogenic Assay

The colony formation of pancreatic cancer cells was evaluated using a crystal violet stain. The cells were cultivated for 96 h with or without treatment with siAHR/siELAVL1/GEM. After treatment, the cells were detached by trypsin/EDTA, counted, and seeded into 6-well culture plates at concentration of 600 cells/well. After 7 days of growth, formed colonies were fixed with 96% ethanol and stained with crystal violet stain. After staining, crystal violet was removed, and the wells were rinsed with water. Plates were dried at room temperature, the morphology of cells was observed, and colonies were counted under an Olympus IX71 phase-contrast microscope (Olympus Corporation, Tokyo, Japan). No less than 4 replicates of the experiment were carried out.

### 4.9. Migration Assay

The cells were cultivated for 96 h with or without treatment with siAHR/siELAVL1/GEM. After treatment, the cells were detached by trypsin/EDTA, counted, and seeded into 24-well culture plates at concentration of 2 × 10^5^ cells/well. After 24 h, a scratch was made with a 200 µL pipette tip, and the medium was changed into a fresh medium without FBS. The scratch was observed and photographed under an Olympus IX71 phase-contrast microscope at 0 and 24 h after making the scratch. No less than 3 replicates of the experiment were carried out.

### 4.10. Immunoprecipitation (IP) Assay

Cells were cultivated in 150 cm^2^ flasks until they reached a confluence of 80–90%. The cells were then lysed using Magna RIP (Millipore, Burlington, MA, USA) kit and the IP assay was carried out using ELAVL1 antibody (Ls-C7451, LsBio, Lynnwood, WA, USA) and GAPDH antibody (AM4300, Invitrogen, Carlsbad, CA, USA) as a negative control. The samples for WB and qRT-PCR analysis were collected before and after immunoprecipitation. WB and qRT-PCR assays were performed as described above.

### 4.11. Statistical Analysis

Statistical analysis was performed using GraphPad (version 6.01; GraphPad Software Inc., La Jolla, CA, USA) software. The data are presented as mean ± SD of three or more independent experiments. A nonparametric Mann–Whitney test was used for comparison between groups. Statistical significance was defined as *p* < 0.05.

## 5. Conclusions

Our study revealed that both AHR and ELAVL1 inter-regulate each other, as well as having a role in cell proliferation, migration, and chemoresistance in PDAC cell lines. Notably, the effect of *AHR* silencing appears to be more pronounced than that of *ELAVL1*, and both proteins act through distinct mechanisms. The silencing of *ELAVL1* disrupts the stability of its target mRNAs, resulting in the decreased expression of numerous cytoprotective proteins. In contrast, the silencing of *AHR* diminishes cell migration and proliferation and enhances cell sensitivity to gemcitabine through the AHR-ELAVL1-DCK molecular pathway.

These findings underscore the complex interplay between AHR, ELAVL1, and important cellular processes in PDAC. The differential effects and distinct molecular pathways associated with AHR and ELAVL1 modulation highlight their potential as therapeutic targets for PDAC treatment, particularly in overcoming chemoresistance and inhibiting tumour progression. Further investigations are necessary to fully elucidate the underlying mechanisms and explore the clinical implications of targeting these pathways in PDAC management.

## Figures and Tables

**Figure 1 ijms-24-13155-f001:**
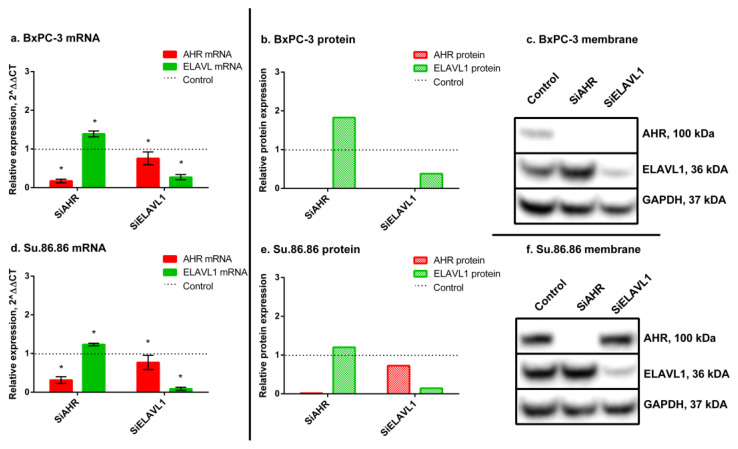
AHR and ELAVL1 qRT-PCR and WB analysis. mRNA and protein expression of AHR and ELAVL1 genes and proteins, after *AHR* or *ELAVL1* silencing by transfection. qRT-PCR N = 3, MEAN ± SD. * *p* < 0.05. WB N = 3; however, only 1 is shown as a representative experiment: (**a**) BxPC-3 qRT-PCR analysis, (**b**) BxPC-3 WB analysis, (**c**) membrane of BxPC-3 WB, (**d**) Su.86.86 qRT-PCR analysis, (**e**) Su.86.86 WB analysis, and (**f**) membrane of Su.86.86 WB.

**Figure 2 ijms-24-13155-f002:**
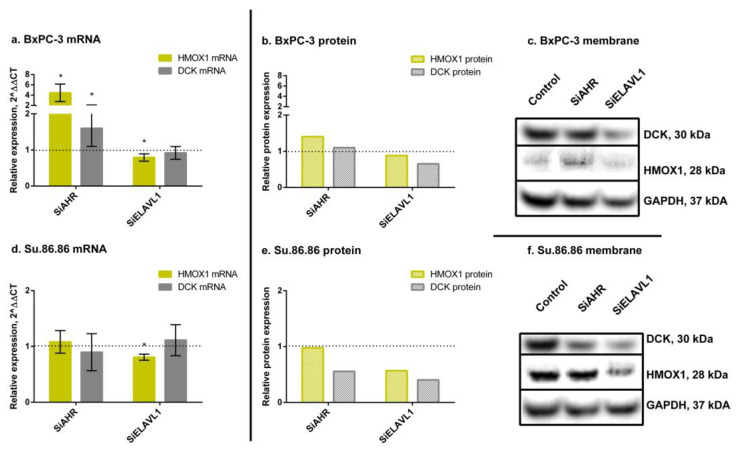
HMOX1 and DCK qRT-PCR and WB analysis. mRNA and protein expression of HMOX1 and DCK genes and proteins, after *AHR* or *ELAVL1* silencing. qRT-PCR N = 3, MEAN ± SD. * *p* < 0.05. WB N = 3; however, only 1 is shown as a representative experiment: (**a**) BxPC-3 qRT-PCR analysis, (**b**) BxPC-3 WB analysis, (**c**) membrane of BxPC-3 WB, (**d**) Su.86.86 qRT-PCR analysis, (**e**) Su.86.86 WB analysis, and (**f**) membrane of Su.86.86 WB.

**Figure 3 ijms-24-13155-f003:**
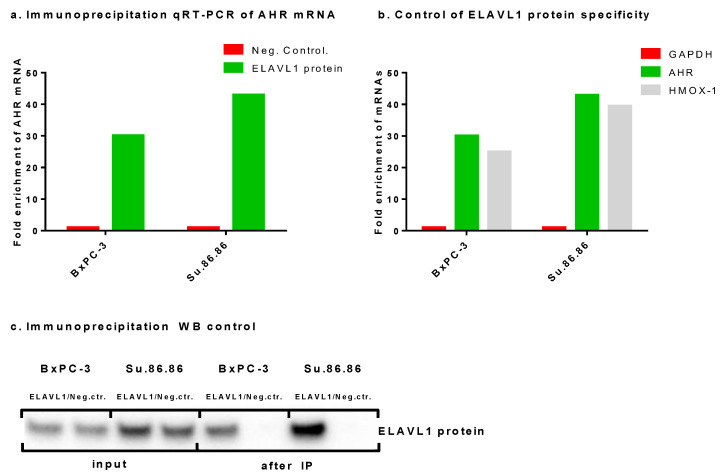
Immunoprecipitation of ELAVL1 protein. IP of ELAVL1 protein showing the ability of the ELAVL1 protein to bind *AHR* mRNA: N = 1 (**a**) qRT-PCR assay showing fold change in *AHR* mRNA when compared with the negative control (*GAPDH*), (**b**) qRT-PCR assay showing *GAPDH* as a negative target mRNA and *AHR* and *HMOX1* as positive mRNA targets for ELAVL1 protein, and (**c**) WB assay showing absence and presence of the ELAVL1 protein after the IP assay.

**Figure 4 ijms-24-13155-f004:**
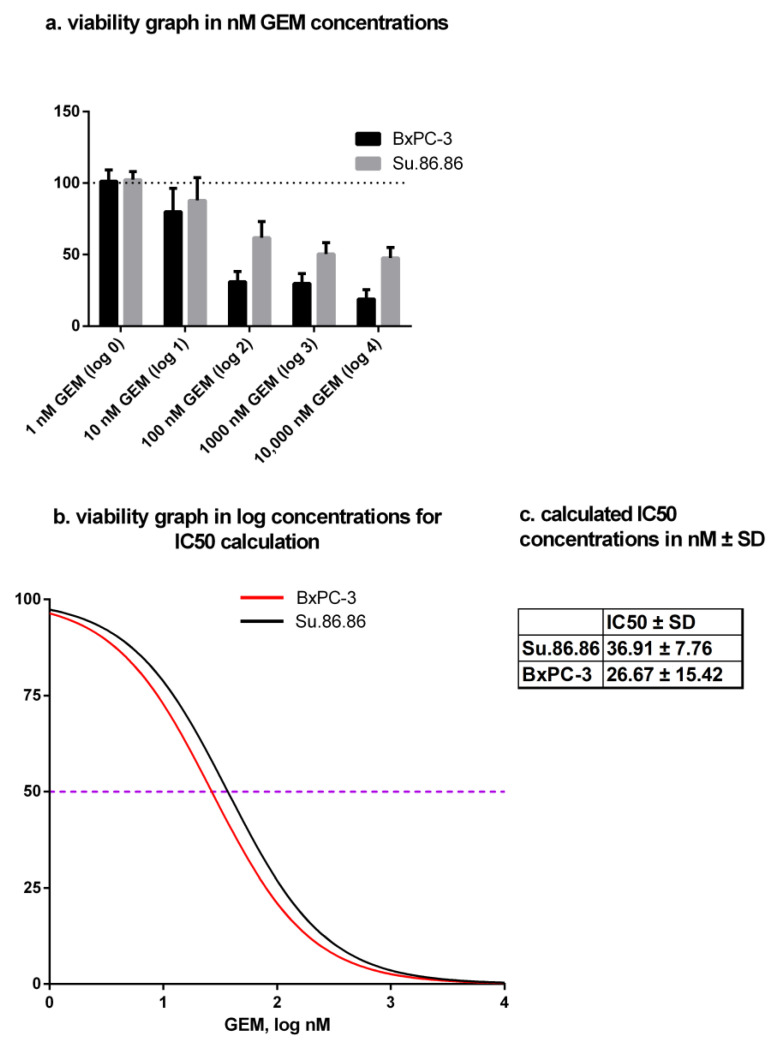
MTT assay and IC50 dose calculation of gemcitabine. IC50 doses in nM ± SD of gemcitabine after 48-h treatment: N ≥ 4 (**a**) viability graph in nM GEM concentration (dotted line–control (100%)), (**b**) viability graph in log concentrations for IC50 calculation (purple dotted line showing 50% viability – IC50), and (**c**) table of numerical value of IC50 doses in nM ± SD.

**Figure 5 ijms-24-13155-f005:**
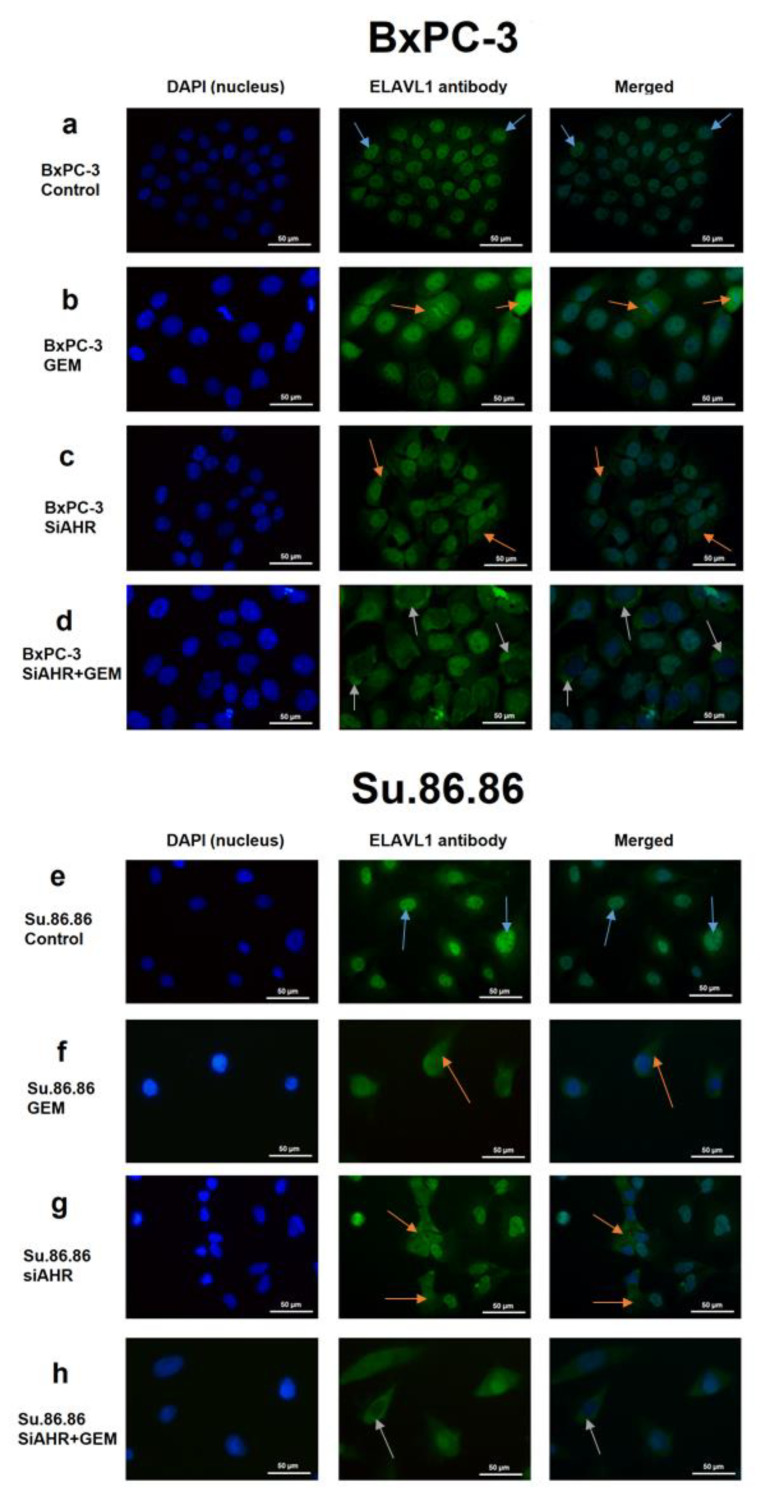
Example photos of PDAC lines ICC with ELAVL1 antibody. Example photos showing the ELAVL1 shift from the nucleus to the cytoplasm after the effect of SiAHR and/or GEM. ICC N = 3; however, only one is being shown as a representative experiment (40× magnification): (**a**) BxPC-3 control, (**b**) BxPC-3 GEM, (**c**) BxPC-3 siAHR, (**d**) BxPC-3 siAHR+GEM, (**e**) Su.86.86 control, (**f**) Su.86.86 GEM, (**g**) Su.86.86 siAHR, and (**h**) Su.86.86 siAHR+GEM. Blue arrows depict ELAVL1 protein being localised predominantly in the nucleus. Orange arrows depict ELAVL1 protein somewhat shifting from the nucleus to the cytoplasm. Grey arrows depict ELAVL1 being localised predominantly in the cytoplasm.

**Figure 6 ijms-24-13155-f006:**
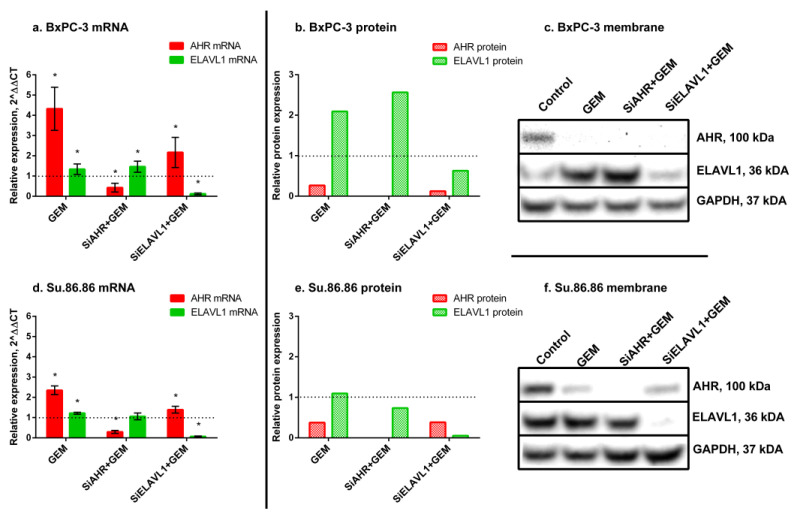
AHR and ELAVL1 qRT-PCR and WB analysis after GEM treatment. mRNA and protein expression of AHR and ELAVL1 genes and proteins, after *AHR* and/or *ELAVL1* silencing by transfection and treatment with IC50 GEM. qRT-PCR N = 3, MEAN ± SD. * *p* < 0.05. WB N = 3; however, only 1 is shown as a representative experiment: (**a**) BxPC-3 qRT-PCR analysis, (**b**) BxPC-3 WB analysis, (**c**) membrane of BxPC-3 WB, (**d**) Su.86.86 qRT-PCR analysis, (**e**) Su.86.86 WB analysis, and (**f**) membrane of Su.86.86 WB.

**Figure 7 ijms-24-13155-f007:**
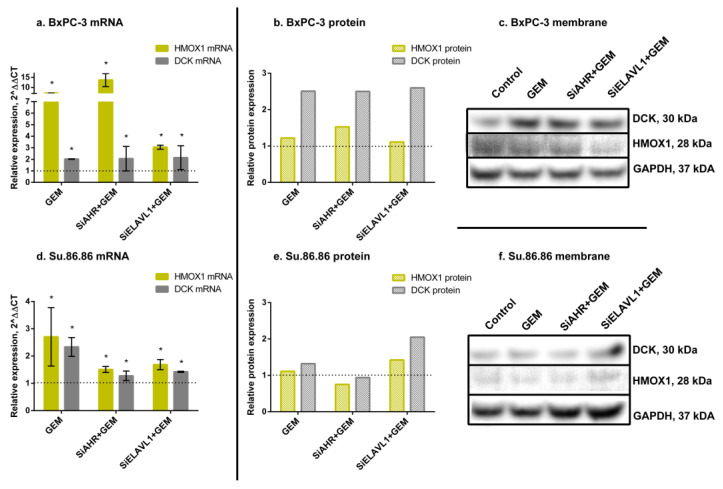
HMOX1 and DCK qRT-PCR and WB analysis after GEM treatment. mRNA and protein expression of HMOX1 and DCK genes and proteins, after *AHR* or *ELAVL1* silencing and treatment with GEM. qRT-PCR N = 3, MEAN ± SD. * *p* < 0.05. WB N = 3; however, only 1 is shown as a representative experiment: (**a**) BxPC-3 qRT-PCR analysis, (**b**) BxPC-3 WB analysis, (**c**) membrane of BxPC-3 WB, (**d**) Su.86.86 qRT-PCR analysis, (**e**) Su.86.86 WB analysis, and (**f**) membrane of Su.86.86 WB.

**Figure 8 ijms-24-13155-f008:**
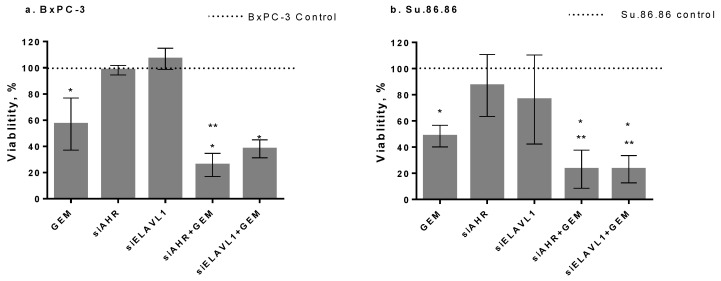
Cell viability analysis by MTT assay. PDAC cell viability after silencing of *AHR* or *ELAVL1* for 24 h and treatment with or without IC50 GEM for 48 h (MTT N = 4): (**a**) BxPC-3 cells and (**b**) Su.86.86 cells. N ≥ 4, MEAN ± SD * *p* < 0.05 when compared with respective control. ** *p* < 0.05 when compared with GEM alone.

**Figure 9 ijms-24-13155-f009:**
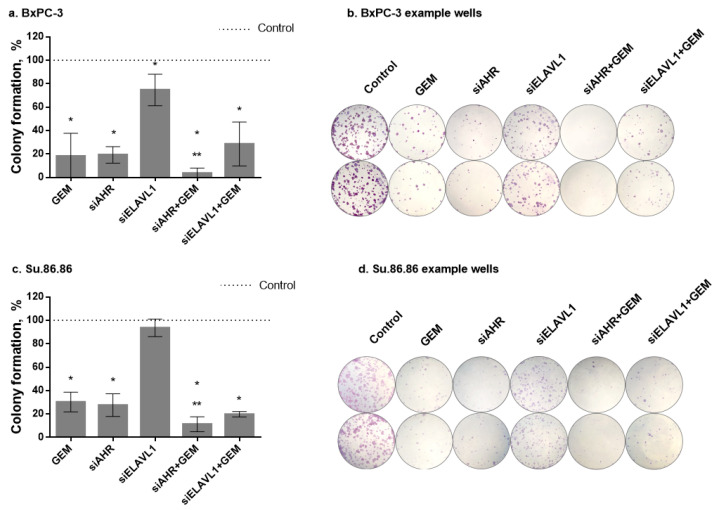
PDAC cell colony formation. Colony formation after silencing *AHR* or *ELAVL1* genes by transfection and/or exposure to GEM. Cells for clonogenic assay were seeded after 24 h transfection and grown for 168 h after seeding: (**a**) BxPC-3 colony formation graph, (**b**) BxPC-3 colony formation example photos, (**c**) Su.86.86 colony formation graph, and (**d**) Su.86.86 colony formation example photos. N ≥ 4, MEAN ± SD * *p* < 0.05 when compared with respective control. ** *p* < 0.05 when compared with GEM alone.

**Figure 10 ijms-24-13155-f010:**
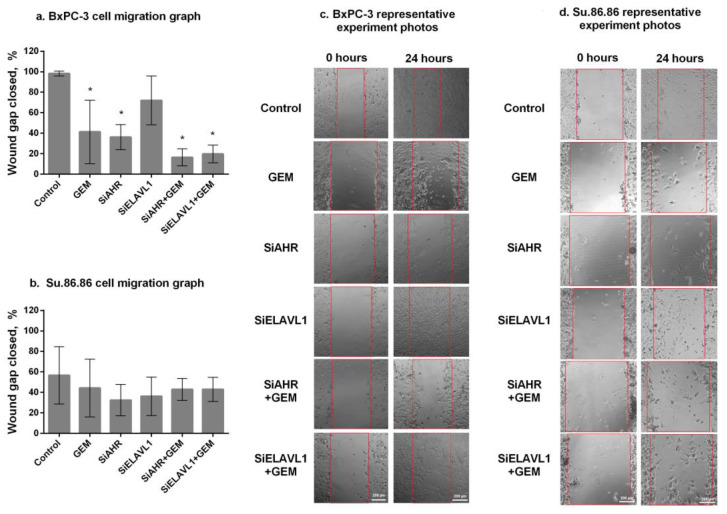
Cell migration by wound healing assay. Two PDAC cell lines showing migratory abilities after the silencing of *AHR* or *ELAVL1* genes and/or effect of gemcitabine: (**a**) BxPC-3 cell line migration graph, (**b**) Su.86.86 cell line migration graph, (**c**) BxPC-3 representative experiment photos (red lines show gap/wound width of photo taken at 0 h), and (**d**) Su.86.86 representative experiment photos (red lines show gap/wound width of photo taken at 0 h). N ≥ 3, MEAN ± SD. * *p* < 0.05 when compared with control.

**Figure 11 ijms-24-13155-f011:**
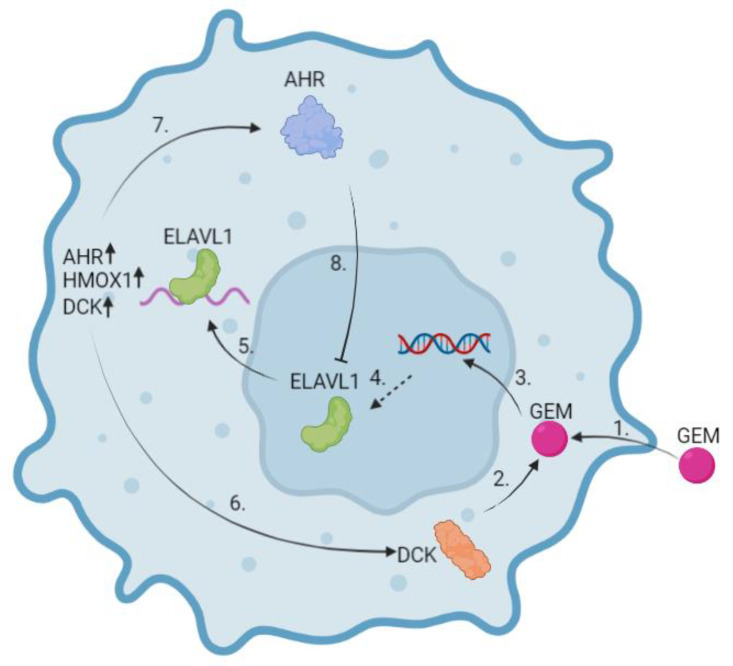
Suggested mechanism of ELAVL1 and AHR influence on gemcitabine effectiveness. In normal conditions, gemcitabine enters the cell through nucleoside transporters (1) where it is phosphorylated to its active state. DCK is the first enzyme to start the phosphorylation mechanism (2). Once GEM is metabolised into its active form, it integrates into the nucleus and stops further cell replication (3). This stress stimulates ELAVL1 (4) to shuttle from the nucleus to the cytoplasm (5). There, ELAVL1 stabilises its target mRNAs, thus promoting the translation of various proteins, including cytoprotective ones such as HMOX1, enzymes such as DCK (6), and the transcription factor AHR (7). Increased DCK synthesis by feedback loop subsequently promotes GEM phosphorylation. However, increased AHR synthesis blocks ELAVL1 from shuttling from the nucleus to the cytoplasm (8). This prevents the proteins of ELAVL1-targeted mRNAs from being overexpressed. (Created in Biorender.com).

## Data Availability

The data presented in this study are available on request from the corresponding author.

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
