# Peer review of "Targeting AHR Increases Pancreatic Cancer Cell Sensitivity to Gemcitabine through the ELAVL1-DCK Pathway"

_ijms, 2023, doi:10.3390/ijms241713155_

Round 1

Reviewer 1 Report

In this manuscript, the authors carry out a study in which they reveal the relationship between the AHR transcription factor and the ELAVL1 antigen in pancreatic ductal adenocarcinoma. There are some aspects that need to be reviewed:

1. The methodology does not specify how many replicates have been carried out for each test, it is necessary to add it to the methodological part and in the captions of figures.

2. The format of the bibliography should be reviewed since the citations sometimes appear in brackets and other times in parentheses, for example, lines 303 and 343.

3. In figure 2.c BxPC3 membrane for HMOX1 it can be seen that the bands correlate with those obtained in the WB gel, but in figure 2.f Su.86.86 membrane for HMOX1 I cannot see the expression, which also makes its quantification difficult. This same thing happens in figure 7 even more pronounced, how reliable is the quantization? How many replicas have been made of each of them? Why do the quantify graphs have no errors?

4. In Figure 4, the Ic50 obtained does not have a standard deviation. How many replicates have been made of each of them? The Ic50 must present SD.

5. Figure 5 must have scale bars both in the images, in the zooms and in the caption.

6. Figure 10 requires a figure showing the images of the culture plates in the migration produced by wound healing assay, at least one representative figure.

Author Response

Thank you for great criticism and suggestions. All the suggestions have been taken into account and answered.

Reviewer 2 Report

The authors of the manuscript entitled " Targeting AHR increases pancreatic cancer cell sensitivity to gemcitabine through the ELAVL1-DCK pathway" where they investigated interaction between AHR and ELAVL1 in the pathogenesis of PDAC in vitro. The authors used siRNA transfection and evaluated their hypothesis through qRTPCR and western blot analysis. The authors also carried out Immune IP to establish the direct binding between ELAVL1 protein and AHR mRNA. They further validated with cell viability assays, clonogenicity and migration assays to show that AHR and ELAVL1 inter regulate with each other. They further showed that by inhibiting AHR expression, PDAC cells became more susceptible to gemcitabine. The manuscript is written however, I have some concerns.

Here are some major concerns I have with this manuscript.

a) with regards to all the western blots provided through out the manuscript, I would suggest that the authors mark the boundary of the western blots. Furthermore, the authors should spend additional time with their blots to match the lanes and the contrast and intensity levels of the bands to make them more convincing. The authors have provided in their supplemental figures the full original blots, however, in case they are using parts of the blot, they should mark them by a solid demarcation and align their lanes properly.

b) With respect to the GAPDH western blot the loading is not even throughout all the lanes. The authors should have estimated the concentration of the proteins to avoid any variance across the lanes. While authors have established the statistical significance based on the intensity of the bands, it is still an established practice in research have a loading control.

c) With regards to the immunofluorescence images that the authors have presented the authors should stick to express the magnified images and highlight the difference they observe by means of arrows, such as the movement of ELAVl1 from the nucleus to the cytoplasm after the effects of siAHR and gemcitabine. In case, the authors want to still show the field of focus  they can show them in the supplementary figures.

d) In case of figure 4 it would provide additional information if the authors had shown the cell viability both in the control cell lines and in the cell lines treated with gemcitabine. The IC50 calculated is additional information.

e) The discussion is well written however, in lines with the reader it would be best if the discussion was precise and concise. In particular, in vitro data is sufficient as a uniform model system. The data provides a stepping stone for in vivo analysis and can men mentioned as beyond the scope of the manuscript. The authors could also mention that such findings needs to be shown in animal models.Furthermore, their discussion should also involve being open to many possibilities and many interaction partners. Limiting to just HMOX1 and DCK as precursors and downstream elements and I strongly the authors should be open to other pathways and interaction partners.

Author Response

Thank you for great criticism and suggestions. All the suggestions have been taken into account and answered. The updated original WB photos have been included in the response.

Round 2

Reviewer 1 Report

The authors have made all suggested changes

Reviewer 2 Report

I thank the authors for considering the suggestions and incorporating them in their manuscript.